# Identification of the genetic basis of pediatric neurogenetic disorders at a tertiary referral hospital in Indonesia: Contribution of whole exome sequencing

Agung Triono[1], Kristy Iskandar[2], Marissa Leviani Hadiyanto[1], Andika Priamas Nugrahanto[1], Kania Diantika[1], Veronica Wulan Wijayanti[1], Elisabeth Siti Herini[1] *

1 Department of Child Health, Faculty of Medicine, Public Health and Nursing, Universitas Gadjah Mada, Dr. Sardjito Hospital, Yogyakarta, Indonesia, 2 Department of Child Health, Faculty of Medicine, Public Health and Nursing, Universitas Gadjah Mada, Academic UGM Hospital, Yogyakarta, Indonesia

* herini_es@ugm.ac.id

## Abstract

### Background

Neurogenetic disorders (NGDs) are complex Mendelian disorders that affect the neurological system. A molecular diagnosis will provide more information about pathophysiology, prognosis, and therapy, including future genetic therapy options. Whole-Exome Sequencing (WES) can rapidly discover the genetic basis in NGDs.

### Objective

The purpose of this study was to assess the WES results and its value in diagnosing pediatric NGDs, especially those with unspecified clinical features.

### Methods

A retrospective chart review was performed from May 2021- February 2023 in Dr. Sardjito General Hospital, a tertiary referral hospital in Yogyakarta, Indonesia. WES proband only was conducted on children aged 0 to 17 years old who met one or more of the following criteria: (1) epileptic encephalopathy and familial epilepsy; (2) complex neurodevelopmental phenotypes; (3) leukodystrophy; (4) movement disorders; and (5) neurocutaneous disorder. The WES was conducted in the certified laboratory, 3Billion, in Seoul, Korea.

### Results

The diagnosis yield of WES in our study was 45% (9/20). We identified nine positive results, including eight pathogenic single nucleotide variants (SNVs) in 8 genes (*KCNQ2, ARSA, UBE3A, IRF2BPL, ATM, MECP2, TSC2,* and *NF1*), and one variant with uncertain significance (VUS) in the *ADK* gene that has not been able to explain the observed clinical features. Of the nine patients with positive WES results, five had missense mutations, three

**Data Availability Statement:** All relevant data are within the paper and its Supporting Information files.

**Funding:** This study was supported by grants from the Ministry of Research and Technology/National Research and Innovation Agency of the Republic of Indonesia to Agung Triono under contract numbers 018/E5/PG.02.00.PT/2022 and Kristy Iskandar under contract numbers 2137/UN1/DITLIT/Dit-Lit/PT.01.03/2023 and. Url of funder website: https://bima.kemdikbud.go.id. The funders had no role in study design, data collection and analysis, decision to publish, or preparation of the manuscript.

**Competing interests:** The authors have declared that no competing interests exist.

**Abbreviations:** ACMG, American College of Medical Genetics; AMP, Association for Molecular Pathology; CMA, chromosomal microarray; CNV, copy number variant; DEE, developmental and epileptic encephalopathy; EEG, Electroencephalogram; ENMG, Electroneuromyography; EMG, Electromyography; MS MLPA, methylation-specific multiplex ligation-dependent probe amplification; MRI, Magnetic Resonance Imaging; NGDs, Neurogenetic disorders; NGS, Next-generation sequencing; SNV, single nucleotide variant; UPD, uniparental paternal disomy; VUS, Variant Uncertain Significance; WES, Whole-Exome Sequencing; WGS, Whole-Genome Sequencing.

frameshift mutations, and one nonsense mutation. Additionally, we identified two suggestive copy number variants (CNVs) in *15q11.2q13.1* and *1p31.3*.

## Conclusions

Whole-Exome Sequencing is an essential diagnostic tool for pediatric NGDs, especially those with unspecified clinical features. It ends multi-year diagnostic odysseys, provides personalized medicine therapy, and optimizes genetic counselling for these families.

## Introduction

Neurogenetic disorders (NGDs) are a complex group of Mendelian disorders that affect the nervous system and have a variety of clinical manifestations [1, 2]. More than half of all genetic anomalies impact the central and/or peripheral nerve systems, either alone or in conjunction with other systems, and the majority of these disorders manifest in childhood [3]. This genetic etiology is especially prevalent in neurologic disorders, such as that causing epilepsy, ataxia, developmental delay, special myopathy, movement disorders, intellectual disability, and others [4]. Many NGDs manifest in early childhood, resulting in early death or lifelong disability. The prevalence of NGDs is about 1 in 1,100 of the general population who experience monogenic neurological disorders [5]. Mutations of a gene can lead to different phenotypes, whereas several other genes can cause the same phenotype [4]. Furthermore, these diseases represent an enormous burden on the affected individuals, including families and communities [6].

The science of pediatric neurogenetics has rapidly developed in the last decade. The development of molecular diagnostic technology reports gene mutations responsible for neurological disorders from various parts of the world [3]. Obtaining a specific molecular diagnosis will provide additional information regarding the pathophysiology, prognosis, and therapy, including genetic therapy opportunities in the future [7]. Next-generation sequencing (NGS) is a new DNA sequencing technique that can sequence a large number of genes, including targeted panel sequencing, whole-exome sequencing (WES) and whole-genome sequencing (WGS). NGS has the capacity to detect several mutations, including de novo, novel, and familial variants [8]. WES focuses on protein-coding sections of the genome, which make up about 1% to 2% of the genome yet contain 85% of mutations with a substantial influence on phenotype [9].

The application of WES technology in research and diagnostic facilities has resulted in the quick discovery of the genetic basis in NGDs. Establishing a genetic etiology in NGDs can also be useful in future planning for the patient and family since it can provide information on prognosis and gene-specific medicines, "precision medicine" [10]. To the best of our knowledge, no research has ever been conducted in our country on the use of WES examination for the basic molecular diagnosis of NGDs. We conducted a retrospective chart review of 20 unrelated individuals with NGDs to assess the WES examination as a diagnostic tool for pediatric NGDs, especially those with unspecified clinical features in routine practice by a tertiary referral hospital in a developing country. All participants had undergone several clinical evaluations and diagnostic tests over the years.

## Methods

### Clinical samples

A retrospective chart review of 20 children with NGDs was performed in the Neuropediatric Clinic at a tertiary regional hospital in Yogyakarta, Indonesia, from May 2021 to February

2023. These patients were chosen as participants for this study based on the presence of NGD symptoms with ages between 0 and 17 years. Inclusion criteria were: (1) Epileptic encephalopathy and familial epilepsy; (2) Complex neurodevelopmental phenotypes, such as microcephaly, and macrocephaly, with unique features of movement disorder and/or epilepsy; (3) Leukodystrophy; (4) Movement disorders, includes familial ataxia, paroxysmal dyskinesia, and/or choreoathetosis with a history of cerebral palsy or epilepsy, early onset dystonia, familial myoclonus, complex movement disorder; and (5) Neurocutaneous disorders. Patients were included if the pediatric neurologists had a strong suspicion that the patient had a genetic disease that met the inclusion criteria. We collected detailed demographic, perinatal and family history, disease progression characteristics, developmental status, comorbidities, and relevant clinical testing such as brain magnetic resonance imaging (MRI), and electroencephalography (EEG). Experienced pediatric neurologists assessed the clinical seizure semiology, EEG results, and brain imaging reports of the patients. All patients' exomes were sequenced and analyzed as probands after getting the patients' or guardians' informed consent. According to the American College of Medical Genetics (ACMG) guidelines (v2.0) [11], patients and/or their legal guardians were informed of the potential revelation of secondary medical outcomes and offered the opportunity to receive the data. If the proband's WES results were positive and the parents were willing to search for the genetic disorder's inheritance, the examination continued to Sanger sequencing of the parents and siblings. The WES and Sanger sequencing were performed in an accredited laboratory, 3Billion, Seoul, Korea. This study was approved by the Medical and Health Research Ethics Committee of the Faculty of Medicine, Public Health and Nursing at Universitas Gadjah Mada, Yogyakarta, Indonesia (K.E./0550/06/2020).

## Whole exome sequencing

Genomic DNA was extracted from each patient's blood samples. The xGen Exome Research Panel v2 was used for exome capture, and the NovaSeq 6000 was used for sequencing. About 99.0% of the targeted bases had a depth of 20x. A total of 69,490 single nucleotide variants (SNV) were identified, and 11,951 small insertions and deletions (indel). On request, the entire variant list is available without annotation or clinical interpretation. Despite insufficient coverage across 1% of the bases, these metrics are consistent with high-quality exome sequencing data and are deemed sufficient for analysis. The analysis of raw genome sequencing data, including alignment to the GRCh37/hg19 human reference genome, variant calling, and annotation, was conducted with open-source bioinformatics tools and custom software [12].

## Variant interpretation

Variant interpretation was performed using EVIDENCE [12], an in-house software developed to prioritize variants based on the guidelines recommended by the ACMG and the AMP [13] in the context of the patient's phenotype, relevant family history and previous test results provided by the ordering physician [11]. The databases that are automatically updated consist of public databases, in-house variant databases, and manually curated literature databases. At the time of variant interpretation, only variants deemed clinically significant and relevant to the patient's primary clinical indications will be reported. Parental and sibling samples (when available) were also tested using Sanger sequencing to confirm clinically relevant variants with proband. The results are interpreted based on the ACMG and AMP standards utilizing a five-tier terminology system called "pathogenic," "likely pathogenic," "variant uncertain significance (VUS)," "likely benign," and "benign" to describe variants identified in genes that cause Mendelian disorders [13].

## Results

### Characteristics of the patients

Twelve males and eight females were included in the study, with the age of presentation ranging from 0 to 17 years. The mean age at onset was 2.45 years ± 3.23 years and the mean age at genetic testing was 6.57 years ± 5.47 years. Table 1 summarizes the demographic and clinical information of the patients. All patients had various neurological manifestations such as: intractable seizure, developmental delay, intellectual delay, regression of milestones, facial dysmorphic, visual impairment, hearing abnormalities, abnormal movement, abnormal gait, abnormal neuroimaging and EEG. Eight patients had primary features of developmental and epileptic encephalopathy (DEE), two patients with leukodystrophy, four patients with a complex neurodevelopmental phenotype, four patients with movement disorders, and two with neurocutaneous disorders, based on the phenotypic evaluation of the pediatric neurologist.

Brain MRI and/or computerized tomography (CT) scans were done on almost all patients (16 out of 20 patients), while the Electroencephalogram (EEG), Electroneuromyography (ENMG) and Electromyography (EMG) were performed according to medical indications. The most common neuroradiological findings were cerebral atrophy (45%), ventriculomegaly (30%), hippocampal atrophy (15%), and hydrocephalus (10%). EEG was performed on 12 of 20 patients, and most had epileptiform discharge with abnormal diffuse irritative.

### Exome sequencing result

Overall, the diagnosis yield in our study was 45% (9/20) (Fig 1). We identified nine positive results, including eight pathogenic single nucleotide variants (SNVs) in 8 genes (*KCNQ2*, *ARSA*, *UBE3A*, *IRF2BPL*, *ATM*, *MECP*, *TSC2*, and *NF1*), one VUS SNVs in the *ADK* gene that has not been able to explain the observed clinical characteristics (Picture 1). All of them are variants that have been published before. Classifications for pathogenic and VUS WES-positive results were identified, but no likely pathogenic, benign, or likely benign variants were identified in this study. Based on the ACMG and AMP five-tier terminology system, we were able to identify pathogenic and VUS variants in this study, but none of those were likely pathogenic, benign, or likely benign. Of the 9 patients with positive WES results, 5 had missense mutations, 3 frameshift mutations, and 1 nonsense mutation. Additionally, we identified two suggestive copy number variants (CNVs) in 15q11.2q13.1 and 1p31.3. In patients with suggestive CNV results, we have not continued further examinations such as chromosomal microarray (CMA). The remaining 11 patients in our study had negative WES results. The average age at onset in the WES-positive group was 3.37 years± 3.36 years, and age at genetic testing was 8.55 years ± 5.65 years. The detailed information about genetic findings of our patients are summarized in Table 2.

We confirmed positive WES results using Sanger sequencing in only 5 of 9 patients, namely patients #1, #9, #16, #19, and #20. Then, parental Sanger sequencing was performed on only three patients with pathogenic SNV (patients #16, #19, and #20) in order to determine inheritance patterns, as the other parents refused to track inheritance. The Sanger sequencing of the parents of patients #19 and #20 was negative, so it can be concluded that the variants in these two patients were de novo. Consequently, we did not continue Sanger sequencing examination of their siblings. Meanwhile, the genetic variant of patient #16 had autosomal recessive inheritance was inherited from asymptomatic parents and patient #16's sibling also underwent Sanger sequencing. Sanger sequencing revealed that the patient (II-2), her sibling (II-1) and her parents (I-1 and I-2) carried the compound heterozygotes of a known nonsense mutation a nonsense mutation at the same position c.8373C>A (p. Tyr2791Ter) in the ATM gene

**Table 1. The demographic and clinical features of patients selected for WES.**

| Patient ID (#) | Primary disease classification | Gender | Familial recurrence | Age at onset | Age at testing | Clinical manifestation | Neuroimaging findings | Other neuroradiologic test |
|---|---|---|---|---|---|---|---|---|
| #1 | DEE | M | - | 1 day | 2 years | Intractable seizure, global developmental delay, cerebral palsy, spastic tetraparesis | Cerebral atrophy, ventriculomegaly, and subdural hygroma | EEG: Abnormal epileptiform |
| #2 | DEE | M | - | 4 years | 15 years | intractable seizure, intellectual disability, Generalized tonic clonic, lower extremity hypertonia, stereotypic movement (clapping hands and head), history of prematurity and low birth weight | Cerebral atrophy, hydrocephalus communicans, hippocampal atrophy, ventriculomegaly | EEG: abnormally irritating with diffuse epileptiform discharge |
| #3 | DEE | M | History of twin with the same symptoms who died. | 2 years | 7 years | intractable seizure, generalized tonic-clonic seizures, cerebral palsy, spastic tetraparesis, speech delay, severe neural hearing loss | N/A | EEG: epileptiform discharge with burst suppression |
| #4 | DEE | M | - | 5 years | 12 years | intractable seizure, global developmental delay generalized tonic-clonic seizures, neurodevelopmental regression, intellectual disability | Cerebral edema, microcalcifications, and cerebritis | EEG: epileptiform discharge with abnormal diffuse irritative |
| #5 | DEE | M | - | 2 day | 1 year | intractable seizure, global developmental delay, focal seizures, strabismus, congenital Hypothyroid, Hypermetropia, cerebral atrophy, infantile spasm | Cerebral atrophy, hippocampal atrophy | EEG: epileptiform discharge with abnormal diffuse irritative and burst suppression |
| #6 | DEE | M | - | 0 day | 0.5 year | Intractable seizure, global developmental delay, hypertonia, Dysmorphic facies | Lissencephaly, Microcephaly, Ventriculomegaly, Bilateral enlargement of the subarachnoid space in the frontotemporoparietal region | N/A |
| #7 | DEE | M | - | 3 years | 8 years | speech delay, generalized tonic-clonic seizures, absence seizure | N/A | EEG: Abnormal epileptiform |
| #8 | DEE | F | - | 1 day | 1.5 years | Infantile spasm, intractable seizure, cerebral atrophy, microcephaly | microcephaly with cerebral atrophy and ventriculomegaly | EEG: epileptiform discharge with abnormal diffuse irritative |
| #9 | leukodystrophy | F | - | 1.5 years | 4 years | Regression of development, generalized tonic clonic seizure and spasticity | white matter changes extensive bihemisphere cerebri | N/A |
| #10 | leukodystrophy | F | - | 4 day | 2 years | Intractable seizure, Focal tonic seizure, global developmental delay, laryngomalacia | Cerebral atrophy, ventriculomegaly, leukodystrophy | EEG: epileptiform discharge with abnormal diffuse irritative |
| #11 | complex neurodevelopmental | M | - | 0 day | 0.5 year | macrocephaly, hypotonia generalisata and facial dysmorphic in the form of frontal bossing, wide forehead, drooping ears, high palatal arch, micrognathia | Cerebral atrophy, ventriculomegaly, and hypoplasia of the corpus callosum | N/A |

*(Continued)*

**Table 1.** (Continued)

| # | | Sex | | | | | | |
|---|---|---|---|---|---|---|---|---|
| #12 | complex neurodevelopmental | M | - | 0 day | 2 years | Coloboma palpebra, nevus sebaceous, infantile spasm, Generalized tonic clonic, intractable seizure, sensorineural hearing loss, global developmental delay | Cerebral atrophy, hippocampal atrophy, communicating hydrocephalus and periventricular cyst | EEG: modified hypsarrhythmia |
| #13 | complex neurodevelopmental | F | History of brother with CP, and sister with speech delay | 3 months | 5 years | Speech delay, Attention Deficit Hyperactivity Disorder | N/A | N/A |
| #14 | complex neurodevelopmental | F | - | 6 months | 4 years | movement and balance disorder, frequent smiling, apparent happy demeanor, speech impairment, absence of seizure | microcephaly, corpus callosum dysgenesis, and heterotopia grey matter on the bilateral lateral ventricle | EEG: abnormal pattern during sleep with medium amplitude rhythmic 2-3c/s |
| #15 | movement disorder | M | - | 5 years | 10 years | Progressive ataxia, Drooling, Slurred speech, Muscle flaccidity | Pathological hyperintensity on FLAIR sequence in bilateral frontal lobes, bilateral basal ganglia, and hippocampus, particularly the left hippocampus. | EMG: myopathy ENMG: motor and sensory results were within normal limits. |
| #16 | movement disorder | F | History of older sister with movement disorder | 6 years | 9 year | Difficulty walking, dysarthria, telangiectasia | Cerebellum atrophy | N/A |
| #17 | movement disorder | M | - | 10 years | 12 years | Dextroscoliosis thoracolumbalis, uncontrolled movement, microcephaly | Bilateral temporal lobe atrophy | N/A |
| #18 | movement disorder | F | - | 2 years | 3 years | Global developmental delay, microcephaly | N/A | N/A |
| #19 | neurocutaneous disorder | F | - | 10 years | 17 years | absence and focal epilepsy, borderline intellectual functioning, organic psychosis, multiple hypomelanotic maculae, angiofibroma, and shagreen patch | Cerebral atrophy, cortical and subcortical tubers in the parietal and occipital lobes | EEG: diffuse epileptiform abnormalities with focus on the left hemisphere |
| #20 | neurocutaneous disorder | M | - | 3 months | 16 years | Generalized tonic clonic seizure, tetraparese spastic, microcephaly cafe au lait spots, Freckling in the armpits or groin area, Lisch nodules, neurofibromas | Cerebral atrophy | EEG: hypsarrthymia, generalizes periodic epileptiform discharge |

DEE: developmental and epileptic encephalopathy; M: Male; F: female; CP: cerebral palsy; N/A: not available; FLAIR: fluid-attenuated inversion recovery; EEG: Electroencephalogram; EMG: Electromyography; ENMG: Electroneuromyography

(Fig 2). The patient and her sibling were affected by homozygous mutations as a result of receiving heterozygous traits from both parents. The patient's 16-year-old brother displayed all the symptoms of ataxia but with a more severe clinical progression. He normally developed until the age of eight, and the symptoms of typical ataxia appeared at nine years old. Both eyes had severe conjunctival hyperemia, with difficulty seeing objects on the right, indicating oculo-motor apraxia. Deep tendon reflexes were not found, and the upper and lower extremities muscle strength were grade III and grade II, respectively. The brain CT scan also revealed cere-bellar atrophy.

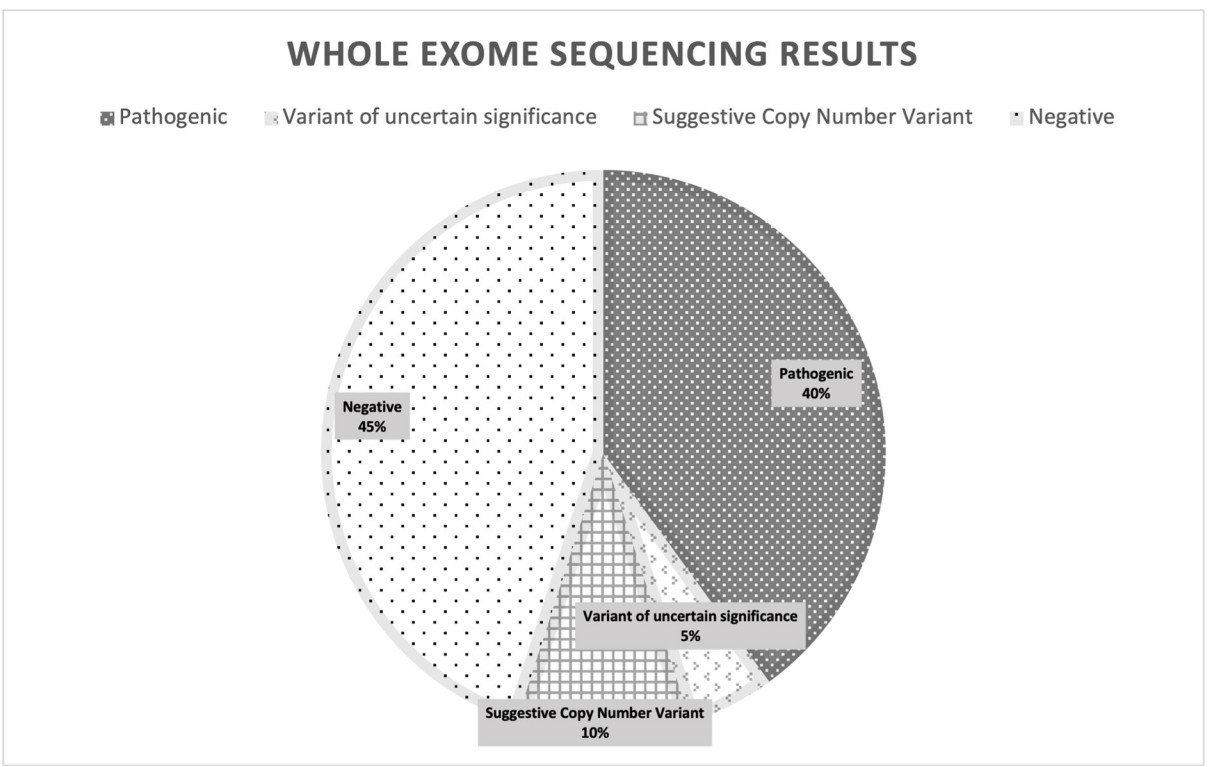

**Fig 1. Whole-exome sequencing result.**

One patient (#4) had *ADK* gene mutation classified as VUS whose phenotypic features did not fully match the genotype. The patient was referred to our hospital at the age of 11 years with an intractable seizure with the onset of seizure at five years old and developmental regression of difficulty maintaining head position (truncal hypotonia). The physical examination revealed normocephalic, no facial dysmorphic, hypertonic, normal deep reflexes, and no pathological reflexes. The testing for serum methionine, S-adenosylmethionine (AdoMet), S-adenosylhomocysteine (AdoHcy), and homocysteine has not been commonly utilized in our country. The *ADK* gene mutations are reported to have a phenotype of global developmental delay, seizures, dysmorphic face, and biochemical anomalies, such as hypermethioninemia with increased levels of AdoMet and AdoHcy, with normal homocysteine.

There were two patients with suggestive CNV. Patient #3, a boy with a heterozygous deletion finding, was identified at 15q11.2q13.1. Deletion at this genomic region is associated with autosomal dominant Prader-Willi syndrome and Angelman syndrome. The heterozygous deletion 15:(?_23684892)_(28544662_?) at 15q11.2q13.1 spans across 14 genes (*GOLGA6L2*, *MKRN3*, *MAGEL2*, *NDN*, *NPAP1*, *SNURF*, *SNRPN*, *UBE3A*, *ATP10A*, *GABRB3*, *GABRA5*, *GABRG3*, *OCA2*, and *HERC2*). He was born after 33 weeks of gestation along with his twin brother, and both were admitted to the Neonatal Incentive Care Unit (NICU) for 10 days due to neonatal asphyxia. This patient's twin brother died at the age of 10 months with intractable seizure and global developmental delay. The patient had developmental delay, ataxia of gait, speech impairment, apparent happy demeanor, microcephaly, and abnormal EEG. These clinical features were consistently found in patients with Angelman syndrome based on the consensus criteria by Williams et al. [14].

Patient #11, a boy with a heterozygous deletion of 6.1 Mb in length in the region of chromosome 1p31.3 leading to Chromosomal Deletion Syndrome 1p32-p31 (OMIM #613735). This

**Table 2. Summary of patients with established molecular diagnosis by WES.**

| Patient ID-(sex) | Primary disease classification | Variant interpretation | Family history | Phenotype | Genes | genomic position (GRCh37) | cDNA changes | Protein changes | Name of disease (OMIM) | Zygosity (Inheritance) |
|---|---|---|---|---|---|---|---|---|---|---|
| #1-M | DEE | Pathogenic | | Intractable seizure, global developmental delay, cerebral palsy, spastic tetraparesis | KCNQ2 | 20-62071010-C-T | NM_172107.4: c.868G>A (missense mutation) | NP_742105.1:p. Gly290Ser | Epileptic encephalopathy, early infantile, (613720) | Heterozygous (unknown) |
| #9-F | leukodystrophy | Pathogenic | | Regression of development, generalized tonic clonic seizure and spasticity | ARSA | 22-51065363-A-AG | NM_000487.6: c.582dup (frameshift mutation) | NP_000478.3:p. Trp195LeufsTer15 | Metachromatic leukodystrophy (250100) | Heterozygous (unknown) |
| #14-F | complex neurodevelopmental | Pathogenic | + | Movement and balance disorder, frequent smiling, apparent happy demeanor, speech impairment, absence seizure | UBE3A | 15-25615817-G-A | NM_000462.5: c.1513C>T (frameshift mutation) | NP_000453.2:p. Arg505Ter | Angelman Syndrome (105830) | Heterozygous (unknown) |
| #15-M | movement disorder | Pathogenic | | Progressive ataxia, Drooling, Slurred speech, Muscle flaccidity | IRF2BPL | 14-77493637-G-A | NM_024496.4: c.499C>T (missense mutation) | NP_078772.1:p. Gln167Ter | Neurodevelopmental disorder with regression, abnormal movements, loss of speech, and seizures (618088) | Heterozygous (unknown) |
| #16-F | movement disorder | Pathogenic | + | Ataxia, dysarthria, telangiectasia | ATM | 11-108214053-C-A | NM_000051.4: c.8373C>A (stop gained-nonsense mutation) | NP_000042.3:p. Tyr2791Ter | Ataxia-telangiectasia syndrome (208900) | Homozygous (autosomal recessive) |
| #18-F | movement disorder | Pathogenic | | Global developmental delay, microcephaly | MECP2 | X-153296363-G-A | NM_001110792.2: c.952C>T (missense mutation) | NP_001104262.1:p. Arg318Cys | X-linked Rett syndrome (OMIM: 312750) | Heterozygous (unknown) |
| #19-F | neurocutaneous disorder | Pathogenic | | Absence and focal epilepsy, borderline intellectual functioning, organic psychosis, multiple hypomelanotic maculae, angiofibroma, and shagreen patch | TSC2 | 16-2137898-C-T | NM_000548.5: c.5024C>T (missense mutation) | NP_000539.2:p. Pro1675Leu | Tuberous Sclerosis 2 (61325) | Heterozygous (de novo) |
| #20-M | neurocutaneous disorder | Pathogenic | | Generalized tonic clonic seizure, tetraparese spastic, microcephaly, cafe au lait spots, Freckling, Lisch nodules, neurofibromas | NF1 | 17-29576097-TC-T | NM_001042492.3: c.4076del (frameshift mutation) | (NP_001035957.1:p. Pro1359LeufsTer19 | Neurofibromatosis, type 1 (162200) | Heterozygous (de novo) |
| #4-M | DEE | VUS | | Intractable seizure, global developmental delay, generalized tonic-clonic seizures, neurodevelopmental regression, intellectual disability | ADK | 10-76153999-T-C | NM_006721.4: c.374T>C (missense mutation) | NP_006712.2:p. Val125Ala | Hypermethioninemia due to adenosine kinase deficiency (614300) | Homozygous (unknown) |

M: male; F: female; DEE: developmental and epileptic encephalopathy; VUS: variant uncertain significance; OMIM: online mendelian inheritance in man

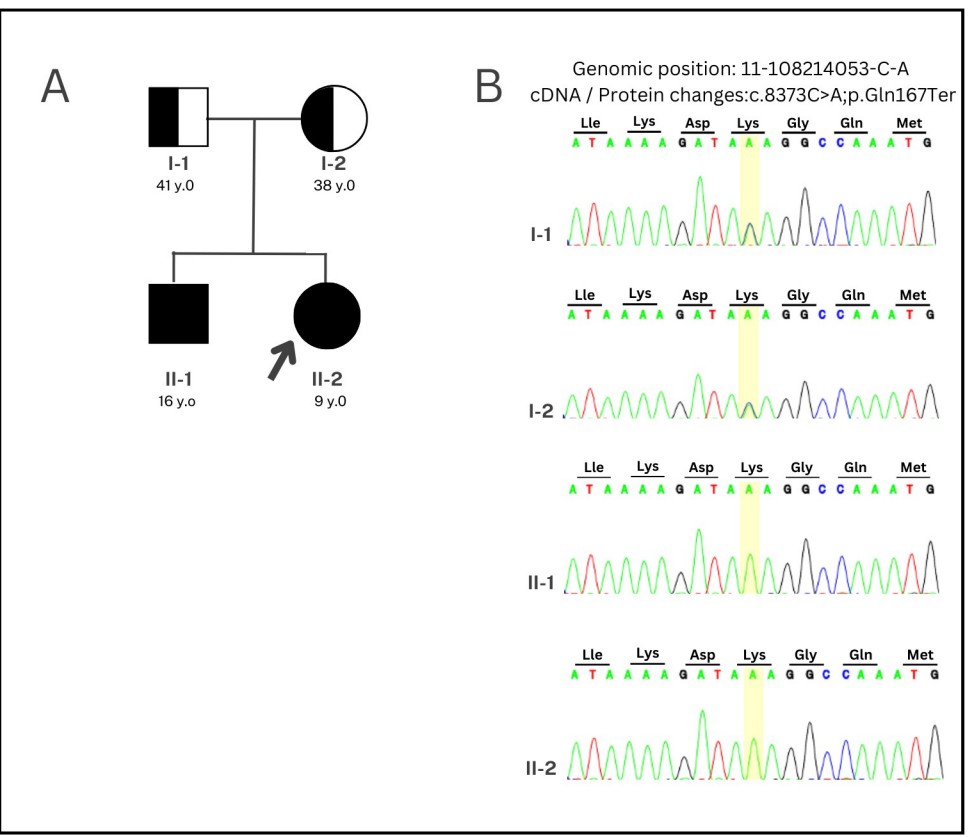

**Fig 2.** (A) Pedigree of patient #16's family and (B) Sanger sequencing results of patient #12 #16 (II-2), her sibling (II-1) and her parents (I-1 and I-2) carried the compound heterozygotes of a known nonsense mutation at the same position c.8373C>A (p. Tyr2791Ter) in the *ATM* gene.

region involves 49 genes including the *Nuclear Factor I/A (NFIA)* gene. The clinical findings of our patient have similar phenotypic features in several other cases of chromosome 1p32-p31 deletion syndrome such as ventriculomegaly, agenesis of the corpus callosum, facial dysmorphic, developmental delay, inguinal hernia, skeletal abnormalities, and urinary system defects. The phenotypic features of this syndrome vary depending on the size of the deletion region and the genes involved.

## Discussion

The introduction of WES into medicine has altered how physicians perform genetic etiology tracing in patients with suspected genetic disorders. WES has recently entered the field of clinical Mendelian disorder diagnosis, with success in diagnosing rare and heterogeneous genetic disorders. We aim to promote the WES examination as a diagnostic tool for pediatric patients with NGDs, especially those with unspecified clinical features. It decreases the time these patients must wait for a diagnosis, ends multi-year odysseys, influences their medical management, and optimizes genetic counseling for these families [15].

In our study, there were positive findings in 45% of patients (9 out of 20), and 13% 10% (2 out of 20) had results suggestive of CNV. Overall, 8 patients with pathogenic results and two patients with suggestive CNV had genetically and phenotypically consistent findings, while one patient with VUS had have an unrelated genotypic phenotype finding. A study in Saudi Arabia reported that 77% (20/26) of pediatric patients with various neurological disorders had

positive WES results [16]. According to a French study, the rate of WES diagnosis reached 32.5% (14/43) in patients aged 2–40 years with intellectual disability (ID) and DEE [17]. A larger sample of 125 pediatric patients in a Dubai study revealed WES had a 40% diagnostic yield in patients with DEE, other focal and generalized epilepsy [18]. In a study of 2,000 children in the United States from different ethnic regions with nervous system dysfunction and developmental delay, WES was able to achieve a basic molecular diagnosis in 25.2% of instances, with 58% of these cases including new mutations [19]. Diagnostic yield for WES varies in studies around the world and is dependent on the size of the WES sample. The diagnostic yield of the WES results in this study, which used a small sample size, was comparable to that of other studies. However, this study still cannot describe the entire population.

The mean age of onset of NGDs in our study was 2.45 ± 3.23 years, while in pediatric movement disorder patients in South Korea, it was 1.8 ± 3.0 years [7]. The mean age of genetic testing in our study was 6.57 ± 5.47 years, while in South Korea, it was 5.3 ± 5.2 years, revealing a delay of around 5 years between age of onset and genetic testing [7]. A study in Argentina even reported that the average time elapsed from the onset of symptoms to WES is 11 years [15]. The delay in genetic diagnosis occurs due to genetic and phenotypic heterogeneity of neurogenetic diseases that forces patients, families and even doctors to undertake a "diagnostic odyssey" [15, 17]. In addition, the WES examination has only gained popularity in our country, Indonesia, during the past 5 years, making it difficult for us to conduct direct genetic testing on patients with atypical symptoms.

WES has a direct impact on the patient' therapy. Although sometimes specific therapy is not available for certain genetic defects but prevention of disease progression and future clinical features can be predicted. For, example, patient number #16 had a positive ATM mutation classified as pathogenic and diagnosed with ataxia telangiectasia (A-T) at nine years old. A-T is a multisystem disorder characterized by progressive neurological impairment, immunodeficiency, chromosomal instability, susceptibility to cancer, and ocular and cutaneous telangiectasia [20]. Evidence from the cerebellum of ATM-deficient mice suggests that oxidative stress may contribute to neuronal abnormalities in A-T. Antioxidant therapy for patients with A-T should be beneficial, given evidence of both diminished antioxidant capacity and elevated levels of oxidative stress in A-T cells [21]. Furthermore, there is also evidence that rehabilitation improves mobility, and balance in people with genetic degenerative ataxia [22]. This patient was given antioxidants (Vitamin A, B1, B2, B3, and B6) and a program of balance and strength exercises with a pediatric physical therapist to prevent A-T progression. Our goal is to prevent rapid progression so that patients can continue to walk and live independently.

WES could reveal a rare illness variant that cannot be detected by conventional genetic tests. WES was used to diagnose Patient #14, a two-year-and-ten-month-old child with Angelman syndrome (AS) due to a point mutation in exon 9 of the *UBE3A* gene. This case report was already published [23]. AS is linked to mutations in the *UBE3A* gene, which is maternally inherited; however, in the case of our patient, the parents refused genetic testing. Several genetic tests, such as single nucleotide polymorphism arrays and methylation-specific multiplex ligation-dependent probe amplification (MS MLPA), are routinely used to screen patient with clinical suspicion of AS. However, these tests cannot detect point mutations; they can only detect maternal deletion, imprinting abnormalities, and paternal uniparental disomy (UPD). While, WES has recently been used to detect point mutation in the *UBE3A* gene [24]. However, WES also has limitations, such as requiring confirmation of other genetic examinations in certain cases. Patient #2, who is still associated with AS, the WES results of this patient showed suggestive CNV with a heterozygous deletion finding identified at 15q11.2q13.1. This finding needs to be confirmed by another appropriate test such as FISH, MLPA, qPCR, CMA or WGS.

In this small sample, it is possible to identify genes that cause rare neurogenetic disorders. Our patient #9, a 4 year old girl who was diagnosed with metachromatic leukodystrophy (MLD) (OMIM 250100), a sulfatide storage disease caused by deficient activity of the lysosomal enzyme arylsulfatase A (ARSA). On the basis of the age of symptom onset, there are three basic MLD phenotypes: late-infantile (< 30 months), juvenile (2.5–16 years), and adult (> 16 years). A severe phenotype is characterized by earlier onset of symptoms, accelerated disease progression, and reduced life expectancy. A search on Clinvar yielded only one finding with the same variant as our patient (NM_000487.6:c.582dup; NP_000478.3:p. Trp195LeufsTer15) [25, 26]. Our patient experienced a developmental regression at the age of 18 months, losing the ability to walk, speak, and grasp. The patient also had several generalized tonic-clonic seizures and is currently spastic in all four extremities. Neurologic examination showed positive signs of meningeal irritation and pathological reflexes (Oppenheim, Clonus and Babinski) and hypertonus. Head MRI examination shows extensive white matter changes in both hemisphere cerebri involving the splenium corpus callosum which leads to a leukodystrophy process. Beerepoot et al. reported patients with the same genotype but different phenotypes with 4 years of age of onset (early juvenile), with symptoms of peripheral neuropathy, ataxia and spasticity [26]. This can contribute to data on phenotypic heterogeneity which is expected to reduce the wandering diagnosis of similar patients in the future.

Although WES is comprehensive, its limitations should be recognized when negative findings are obtained, particularly when clinical indications are prominent; a multi-pronged approach is required before concluding the absence of genetic abnormalities; limited ability to detect CNV in a heterozygous state. We summarized some recommendations when finding a negative WES result [17, 27, 28]. These included: 1) Add more detailed clinical phenotypic and family history data to make an accurate differential diagnosis.; 2) Perform an annual bioinformatic reanalysis of the data with a focus on uncommon variants that affect the coding sequence of genes listed in OMIM between analyses; 3) WES can consistently detect only single nucleotide variations and short (20bp) insertions and deletions in the protein-coding exonic regions. If large deletion/duplication, translocation, inversion, low-level mosaicism, or mitochondrial genome variants are suspected, it is advised to conduct tests specialized to detect these variant types.; 4) Testing (WES or Sanger sequencing) of biological parents and/or other family members is recommended to confirm the segregation of the variant(s).

## Conclusions

WES is an important tool for diagnosing NGDs in children, especially those with unspecified clinical symptoms in routine practice. It decreases the time these patients must wait for a diagnosis, ends multi-year odysseys, gets personalized medicine therapy, and optimizes genetic counselling for these families.

## Limitations

Given our limited sample size and sampling based on physicians' decisions, it is not possible to draw definite conclusions and generalizations. Another limitation of this study is that we were unable to confirm the Sanger Sequencing of in all probands with positive WES results and trace the inheritance of genetic variants from the parents and siblings because of our limited resource settings. In the future, a comprehensive prospective study should be conducted to investigate the clinical characteristics that predict favorable genetic outcomes.

## Supporting information

**S1 File. Schematics of WES analysis workflow in 3billion.**
(DOCX)

## Acknowledgments

We would like to thank Gunadi, MD, PhD for his assistance in consulting several patient genetic results. We are grateful for the staff of the Office of Research and Publication, Faculty of Medicine, Public Health and Nursing, Universitas Gadjah Mada for English editing services and assistance in the proofreading and editing process.

## Author Contributions

**Conceptualization:** Agung Triono, Elisabeth Siti Herini.

**Data curation:** Agung Triono, Kristy Iskandar, Elisabeth Siti Herini.

**Formal analysis:** Agung Triono, Kristy Iskandar, Marissa Leviani Hadiyanto, Andika Priamas Nugrahanto, Kania Diantika, Veronica Wulan Wijayanti, Elisabeth Siti Herini.

**Investigation:** Agung Triono, Elisabeth Siti Herini.

**Methodology:** Agung Triono, Kristy Iskandar, Elisabeth Siti Herini.

**Project administration:** Agung Triono, Elisabeth Siti Herini.

**Resources:** Agung Triono, Elisabeth Siti Herini.

**Software:** Agung Triono, Elisabeth Siti Herini.

**Supervision:** Agung Triono, Elisabeth Siti Herini.

**Validation:** Agung Triono, Kristy Iskandar, Marissa Leviani Hadiyanto, Andika Priamas Nugrahanto, Kania Diantika, Veronica Wulan Wijayanti, Elisabeth Siti Herini.

**Visualization:** Agung Triono, Elisabeth Siti Herini.

**Writing – original draft:** Agung Triono, Kristy Iskandar, Marissa Leviani Hadiyanto, Andika Priamas Nugrahanto, Kania Diantika, Veronica Wulan Wijayanti, Elisabeth Siti Herini.

**Writing – review & editing:** Agung Triono, Kristy Iskandar, Marissa Leviani Hadiyanto, Andika Priamas Nugrahanto, Kania Diantika, Veronica Wulan Wijayanti, Elisabeth Siti Herini.

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
