## [Decision Letter · Decision Letter 0]

17 Jul 2023

PONE-D-23-14709Identification of the genetic basis of pediatric neurogenetic disorders at a tertiary referral hospital in Indonesia: contribution of whole exome sequencingPLOS ONE

Dear Dr. Herini,

Thank you for submitting your manuscript to PLOS ONE. After careful consideration, we feel that it has merit but does not fully meet PLOS ONE’s publication criteria as it currently stands. Therefore, we invite you to submit a revised version of the manuscript that addresses the points raised during the review process.

We look forward to receiving your revised manuscript.

Kind regards,

Nejat Mahdieh

Academic Editor

PLOS ONE

Journal Requirements:

Reviewers' comments:

Reviewer's Responses to Questions

**Comments to the Author**

1. Is the manuscript technically sound, and do the data support the conclusions?

Reviewer #1: Yes

Reviewer #2: Partly

2. Has the statistical analysis been performed appropriately and rigorously? 

Reviewer #1: N/A

Reviewer #2: N/A

3. Have the authors made all data underlying the findings in their manuscript fully available?

Reviewer #1: Yes

Reviewer #2: Yes

4. Is the manuscript presented in an intelligible fashion and written in standard English?

Reviewer #1: Yes

Reviewer #2: Yes

5. Review Comments to the Author

Reviewer #1: Dear Authors,

Thank you for the molecular diagnosis work you have stepped forward in your country.

The Authors have genetically evaluated the genetic causes of 20 cases with neurologic disorders by WES. According to their study they have found the causes of 9 patients. And they found 2 CNV.

A few comment to your study:

1. Genetic diagnosis of the patients by WES should be confirmed by different evaluation depending on the cases. Patient 16, 19 and 20 were evaluated only for Sanger sequencing of the parents. ATM gene, TSC2 and NF gene were evaluated in family members. Please indicate the results.

2. Patient 16, page 13, please indicate what you mean by compound heterozygote in parents. Line174.

3. It seems that ATM gene variation was found in two cases (16 and 12) in a family please indicate it in the table. Please clarify.

4. MLD has an AR inheritance. In patient 9 you have only found one of the variants. Please clarify. Is it confirmed in the parents?

5. In patient 3 and patient 11, did you confirm the result for diagnosis?

6. Did the authors confirmed UBE3A gene variant with any other method? As discussed parents refused genetic testing.

Reviewer #2: I would like to congratulate to authors for doing this study. They assessed WES as a genetic diagnostic tool in 20 patients with Neurogenetic disorders (NGDs).

The overall concept of the study is remarkable however; there are some concerns that need to be addressed:

1- Although the writing English is acceptable in general, there are two incomplete sentences (lines 57 and 155) and several grammatical errors (even in the abstract) that need to be resolved. Some of them are highlighted in the attached file.

2- The genetic concepts sometimes were used in an inappropriate way. For example, in the line 172, it is stated that patient #16 had autosomal recessive inheritance. While autosomal recessive inheritance is attributed to the disease, not the patient. Thus the genetic parts of this manuscript need to reviewed by a genetic expert.

3- In the lines 173-175, with stating only one mutation, compound heterozygosity in two sibling and their parents is unclear. Please rewrite this part in a more clear way.

4- Is there any relationship between patient #12 and patient #16? Why patient #12 is not indicated in Table 2?

5- OMIM# 613735 mentioned in the line 209 is related to "Brain malformations with or without urinary tract defects" not the mentioned chromosomal deletion syndrome.

6- In the lines 224 and 225 there are conflicting numbers about number of patients with positive findings (8 vs. 9). Also, how do they get to the 13% suggestive CNVs?

7- In the line 275, other techniques such as FISH, MLPA or even qPCR can be also used in order to confirm deletions in Angelman syndrome.

8- In the discussion limitation of WES for detection of CNVs in heterozygous state should be acknowledged.

9- In the line 301, short (20bp) insertions and deletions should be corrected as WES can detect indels even larger than 20bp.

10- Genome build used for nomenclature and genetic diagnosis should be stated in the method section.

6. PLOS authors have the option to publish the peer review history of their article (what does this mean?). If published, this will include your full peer review and any attached files.

Reviewer #1: No

Reviewer #2: **Yes: **Ali Rashidi-Nezhad

---

## [Author Response · Author response to Decision Letter 0]

1 Oct 2023

RESPONSES TO REVIEWERS AND EDITOR:

In the material below, the editor and reviewer comments are in italics and our responses are in regular type.

REVIEWER 1

1. Genetic diagnosis of the patients by WES should be confirmed by different evaluation depending on the cases. Patient 16, 19 and 20 were evaluated only for Sanger sequencing of the parents. ATM gene, TSC2 and NF gene were evaluated in family members. Please indicate the results.

Response: Thank you for your suggestion. 

- We have added sentence at the results section to clarify the result of Sanger sequencing of the probands, parents and other family member “We confirmed positive WES results using Sanger sequencing in only 5 of 9 patients, namely patients #1, #9, #16, #19, and #20. Then, parental Sanger sequencing was performed on only three patients with pathogenic SNV (patients #16, #19, and #20) in order to determine inheritance patterns, as the other parents refused to track inheritance. The Sanger sequencing of the parents of patients #19 and #20 was negative, so it can be concluded that the variants in these two patients were de novo. Consequently, we did not continue Sanger sequencing examination of their siblings. Meanwhile, patient #16 was inherited from asymptomatic parents and patient #16's sibling also underwent Sanger sequencing.“ RESULT, line 174-181

- What we have stated above is one of the limitations of our study, as stated in the section on limitations: "Another limitation of this study is that we were unable to confirm the Sanger Sequencing of in all probands with positive WES results and trace the inheritance of genetic variants from the parents and siblings because of our limited resource settings” LIMITATIONS, line 325-328

2. Patient 16, page 13, please indicate what you mean by compound heterozygote in parents. Line174.

Response: Thank you for your input, we would like to make a correction Based on the results of Sanger Sequencing on Patient #16 (the proband), the parents, and the sibling, it was determined that all four individuals have genetic variants in the same position. Thus, we can say that patient #16 and her sibling had homozygous mutations as a result of receiving heterozygous traits from both parents, and not compound heterozygous.

 “Sanger sequencing revealed that the patient (II-2), her sibling (II-1) and her parents (I-1 and I-2) carried a nonsense mutation at the same position c.8373C>A (p. Tyr2791Ter) in the ATM gene (Fig 2). The patient and her sibling were affected by homozygous mutations as a result of receiving heterozygous traits from both parents.” RESULTS, line 181-184

3. It seems that ATM gene variation was found in two cases (16 and 12) in a family please indicate it in the table. Please clarify.

Response: Thank you for your question. We made a typo in the description of Fig 2 (Figure 2's description says patient#12, but it should be patient#16) and we have already corrected it. 

“Fig 2. (A) Pedigree of patient #16's family and (B) Sanger sequencing results of patient #12 #16 (II-2), her sibling (II-1) and her parents (I-1 and I-2) carried a nonsense mutation at the same position c.8373C>A (p. Tyr2791Ter) in the ATM gene.” RESULTS, line 193-195

4. MLD has an AR inheritance. In patient 9 you have only found one of the variants. Please clarify. Is it confirmed in the parents?

Response: Thank you for your question. No, we have not performed Sanger tests on the parents of patient #9 to determine whether the variant is inherited or de novo because of our limited resource settings. 

5. In patient 3 and patient 11, did you confirm the result for diagnosis?

Response: Thank you for your question, Unfortunately, a definitive genetic diagnosis cannot be provided at this time because of our limited resource settings. Patients #3 and #11 have CNV, which must be confirmed by WGS or CMA before a conclusive genetic diagnosis can be made. 

6. Did the authors confirmed UBE3A gene variant with any other method? As discussed parents refused genetic testing. 

Response: Thank you for your question. No, we did not perform confirmation tests for the UBE3A variant in patient #14, such as Sanger sequencing or MLPA. This is one of the limitations of our study where not all patients underwent confirmation tests. 

REVIEWER 2

1. Although the writing English is acceptable in general, there are two incomplete sentences (lines 57 and 155) and several grammatical errors that need to be resolved. Some of them are highlighted in the attached file. 

Response: Thank you for the suggestion. 

- This manuscript has been checked for grammatical errors and has been proofread by the Senior Consultant of Academic Writing and English Literacy of Faculty of Medicine, Public Health, and Nursing, Gadjah Mada University. Here we attach the Certificate of Manuscript Editing. 

- We also have corrected incomplete sentences in the manuscript.

a. Before:“ The prevalence of NGDs is about 1 in 1,100 of the general population who experience neurological disorders with a single gene” 

After: “The prevalence of NGDs is about 1 in 1,100 of the general population who experience monogenic neurological disorders.” INTRODUCTION, line 56-58

b. Before: “Classifications for pathogenic and VUS WES-positive results were identified, but no likely pathogenic, benign, or likely benign variants were identified in this study.” 

After: “Based on the ACMG and AMP five-tier terminology system, we were able to identify pathogenic and VUS variants in this study, but none that were likely pathogenic, benign, or likely benign." RESULTS, line 161-163

2. The genetic concepts sometimes were used in an inappropriate way. For example, in the line 172, it is stated that patient #16 had autosomal recessive inheritance. While autosomal recessive inheritance is attributed to the disease, not the patient. Thus the genetic parts of this manuscript need to reviewed by a genetic expert.

- Response: Thank you for the suggestion and we strongly agree with your suggestion. We have corrected the sentence “Meanwhile, the genetic variant of patient #16 had autosomal recessive inheritance was inherited from asymptomatic parents and patient #16's sibling also underwent Sanger sequencing “ RESULT, line 180

3. In the lines 173-175, with stating only one mutation, compound heterozygosity in two sibling and their parents is unclear. Please rewrite this part in a more clear way.

Response: Thank you for your input, we would like to make a correction. Based on the results of Sanger Sequencing on Patient #16 (the proband), the parents, and the sibling, it was determined that all four individuals have genetic variants in the same position. Thus, we can say that patient #16 and her sibling had homozygous mutations as a result of receiving heterozygous traits from both parents, and not compound heterozygous. 

“Sanger sequencing revealed that the patient (II-2), her sibling (II-1) and her parents (I-1 and I-2) carried a nonsense mutation at the same position c.8373C>A (p. Tyr2791Ter) in the ATM gene (Fig 2). The patient and her sibling were affected by homozygous mutations as a result of receiving heterozygous traits from both parents.” RESULTS, line 181-185

4. Is there any relationship between patient #12 and patient #16? Why patient #12 is not indicated in Table 2?

Response: Thank you for your question, Patient #16 and #12 are not related. We made a typo in the description of Fig 2 (Figure 2's description says patient#12, but it should be patient#16) and have already corrected it. “Fig 2. (A) Pedigree of patient #16's family and (B) Sanger sequencing results of patient #12 #16 (II-2), her sibling (II-1) and her parents (I-1 and I-2) carried a known nonsense mutation at the same position c.8373C>A (p. Tyr2791Ter) in the ATM gene.” RESULT, line 193-195

5. OMIM# 613735 mentioned in the line 209 is related to "Brain malformations with or without urinary tract defects" not the mentioned chromosomal deletion syndrome.

Response : Thank you for your response, Based on the OMIM website https://www.omim.org/entry/613735, BRAIN MALFORMATIONS WITH OR WITHOUT URINARY TRACT DEFECTS can also be called CHROMOSOME 1p32-P31 DELETION SYNDROME.

6. In the lines 224 and 225 there are conflicting numbers about number of patients with positive findings (8 vs. 9). Also, how do they get to the 13% suggestive CNVs?

Response : Thank you for your response, We've corrected some misspellings in that 2 sentences. 

“In our study, there were positive findings in 45% of patients (9 out of 20), and 13% 10% (2 out of 20) had results suggestive of CNV. Overall, 8 patients with pathogenic results and two patients with suggestive CNV had genetically and phenotypically consistent findings, while one patient with VUS had have an unrelated genotypic phenotype finding.” DISCUSSION, line 234-237

7. In the line 275, other techniques such as FISH, MLPA or even qPCR can be also used in order to confirm deletions in Angelman syndrome.

Response : Thank you for your response, we agree that FISH, MLPA or even qPCR can also be used in order to confirm deletions in Angelman syndrome. We already corrected the sentence “Patient #2, who is still associated with AS, the WES results of this patient showed suggestive CNV with a heterozygous deletion finding identified at 15q11.2q13.1. This case requires FISH, MLPA, qPCR, CMA or WGS genetic testing to confirm the findings suggestive of CNV.” DISCUSSION, line 283-286

8. In the discussion limitation of WES for detection of CNVs in heterozygous state should be acknowledged.

Response : Thank you for your suggestion, we have added your suggestion to the sentence regarding the limitations of WES “Although WES is comprehensive, its limitations should be recognized when negative findings are obtained, particularly when clinical indications are prominent; a multi-pronged approach is required before concluding the absence of genetic abnormalities; inability to detect CNV in a heterozygous state.” DISCUSSION, line 306-307

9. In the line 301, short (20bp) insertions and deletions should be corrected as WES can detect indels even larger than 20bp.

Response: Thank you for your suggestion. We have already changed the sentence “WES can consistently detect only single nucleotide variations and short (20bp) insertions and deletions in the protein-coding exonic regions.” DISCUSSION, line 311-312

10. Genome build used for nomenclature and genetic diagnosis should be stated in the method section. 

Response: Thank you for your suggestion.

- About the genome build, we have added sentence at the methods section “The analysis of raw genome sequencing data, including alignment to the GRCh37/hg19 human reference genome, variant calling, and annotation, was conducted with open-source bioinformatics tools and custom software.” METHODS, line 115-117

- We have written about genetic diagnosis (variant interpretation) in this study in the method variant interpretation section “ Variant interpretation was performed using EVIDENCE [12], an in-house software developed to prioritize variants based on the guidelines recommended by the ACMG and the AMP [13] in the context of the patient's phenotype, relevant family history and previous test results provided by the ordering physician [11]. The databases that are automatically updated consist of public databases, in-house variant databases, and manually curated literature databases. At the time of variant interpretation, only variants deemed clinically significant and relevant to the patient's primary clinical indications will be reported. “METHODS, line 120-126

---

## [Decision Letter · Decision Letter 1]

6 Oct 2023

Identification of the genetic basis of pediatric neurogenetic disorders at a tertiary referral hospital in Indonesia: contribution of whole exome sequencing

PONE-D-23-14709R1

Dear Dr. Elisabeth Siti Herini, pleased to inform you that your manuscript has been judged scientifically suitable for publication and will be formally accepted for publication once it meets all outstanding technical requirements.

Kind regards,

Nejat Mahdieh

Academic Editor

PLOS ONE

Additional Editor Comments (optional):

Reviewers' comments:

Reviewer's Responses to Questions

**Comments to the Author**

1. If the authors have adequately addressed your comments raised in a previous round of review and you feel that this manuscript is now acceptable for publication, you may indicate that here to bypass the “Comments to the Author” section, enter your conflict of interest statement in the “Confidential to Editor” section, and submit your "Accept" recommendation.

Reviewer #2: (No Response)

2. Is the manuscript technically sound, and do the data support the conclusions?

Reviewer #2: (No Response)

3. Has the statistical analysis been performed appropriately and rigorously? 

Reviewer #2: (No Response)

4. Have the authors made all data underlying the findings in their manuscript fully available?

Reviewer #2: (No Response)

5. Is the manuscript presented in an intelligible fashion and written in standard English?

Reviewer #2: (No Response)

6. Review Comments to the Author

Reviewer #2: I would like to thank the authors for their efforts to improve their manuscript.

There are still 3 points that need to be resolved:

1- In the line 162 "none that" should be replaced by "none of those"

2- In the line 286, it is suggested that this sentence "This case requires FISH, MLPA, qPCR, CMA or WGS genetic testing to confirm the findings suggestive of CNV" to be replaced by "This finding needs to be confirmed by another appropriate test such as FISH, MLPA, qPCR, CMA or WGS".

3- In the line 306 the phrase "inability to detect" to be replaced by "Limited ability to detect"

7. PLOS authors have the option to publish the peer review history of their article (what does this mean?). If published, this will include your full peer review and any attached files.

Reviewer #2: **Yes: **Ali Rashidi-Nezhad

---

## [Editor Report · Acceptance letter]

13 Oct 2023

PONE-D-23-14709R1 

Identification of the genetic basis of pediatric neurogenetic disorders at a tertiary referral hospital in Indonesia: contribution of whole exome sequencing 

Dear Dr. Herini:

I'm pleased to inform you that your manuscript has been deemed suitable for publication in PLOS ONE. Congratulations! Your manuscript is now with our production department. 

Kind regards, 

on behalf of

Dr. Nejat Mahdieh 

Academic Editor

PLOS ONE